# Risks for death after admission to pediatric intensive care (PICU)—A comparison with the general population

Tova Hannegård Hamrin[1,2]*, Staffan Eksborg[1,3]

**1** Pediatric Perioperative Medicine and Intensive Care, Astrid Lindgren Children's Hospital, Karolinska University Hospital Solna, Stockholm, Sweden, **2** Department of Physiology and Pharmacology, Section of Anesthesiology and Intensive Care, Karolinska Institutet, Astrid Lindgren Children's Hospital, Karolinska University Hospital Solna, Stockholm, Sweden, **3** Department of Women's and Children's Health, Childhood Cancer Research Unit, Karolinska Institutet, Astrid Lindgren Children's Hospital, Barnläkemedelsgruppen, Karolinska University Hospital Solna, Stockholm, Sweden

* tova.hannegard.hamrin@ki.se

**Data Availability Statement:** All relevant data are within the paper and its Supporting information files.

## Abstract

### Objective/aim

The aim of the study was to quantify excess mortality in children after admission to a Pediatric Intensive Care Unit (PICU), compared to the age and sex matched general Swedish population.

### Design

Single-center, retrospective cohort study.

### Setting

Registry study of hospital registers, a national population register and Statistics Sweden.

### Patients

Children admitted to a tertiary PICU in Sweden in 2008–2016.

### Interventions

None.

### Main results

In total, 6,487 admissions (4,682 patients) were included in the study. During the study period 444 patients died. Median follow-up time for the entire PICU cohort was 7.2 years (IQR 5.0–9.9 years). Patients were divided into four different age groups (0–28 d, > 28 d -1 yr, > 1–4 yr, and > 4 yr) and four different risk stratification groups [Predicted Death Rate (PDR) intervals: 0–10%, > 10–25%, > 25–50%, and > 50%] at admission. Readmission was seen in 929 (19.8%) patients. The Standardized Mortality Ratios (SMRs) were calculated using the matched Swedish population as a reference group. The SMR for the entire study

**Funding:** Tova Hannegård Hamrin (THH) reported grants from Her Royal Highness Crown Princess Lovisa Foundation: reference id 2021-00670. https://www.kronprinsessanlovisa.se/ The funder had no role in study design, data collection and analysis, decision to publish, or preparation of the manuscript.

**Competing interests:** The authors have declared that no competing interests exist.

group was 49.8 (95% CI: 44.8–55.4). For patients with repeated PICU admissions SMR was 108.0 (95% CI: 91.9–126.9), and after four years 33.9 (95% CI: 23.9–48.0). Patients with a single admission had a SMR of 35.2 (95% CI: 30.5–40.6), and after four years 11.0 (95% CI: 7.0–17.6). The highest SMRs were seen in readmitted children with oncology/hematology (SMR = 358) and neurologic (SMR = 192) diagnosis. Children aged >1–4 years showed the highest SMR (325). In PDR group 0–10% children with repeated PICU admissions (n = 798), had a SMR of 100.

## Conclusion

Compared to the matched Swedish population, SMRs were greatly elevated up to four years after PICU admission, declining from over 100 to 33 for patients with repeated PICU admissions, and from 35 to 11 for patients with a single PICU admission.

## Introduction

Mortality rates in pediatric intensive care units (PICU) have decreased significantly during the last decades [1–3]. The reasons are multifactorial: centralization of intensive care for children to tertiary PICUs, medical and technological advances, dedicated pediatric inter-hospital transport teams, and specialized training in pediatric intensive care are some [4–8]. Children who would previously have died, survive because of these improvements in intensive care. There is also a growing number of PICU patients who would not have been admitted previously due to preexisting complex chronic illnesses [5, 6]. It has been hypothesized that improving intensive care survival and expanded indications for intensive care despite complex chronic disease, contribute to increasing post-PICU mortality. Thus, assessing PICU mortality alone might not reflect the true severity of pediatric critical illness. In a previous study we found that critically ill children in need of transport to a tertiary PICU showed a continued mortality risk after PICU discharge [9].

Studies on outcomes after pediatric intensive care have assessed mortality and morbidity from a wide range of different aspects: certain ICU-diagnoses and patient groups, admission to ICU vs PICU, different follow-up time, etc. [1, 2, 10–16]. Only two studies have compared outcome to a non-ICU, age-matched control population; a Swedish study comparing children treated in PICU and general ICU 1998–2001 and a Finnish study investigating children admitted to PICU or general ICU 2009–2010 [17, 18]. In these two studies, data on the severity of illness at admission was lacking, hence there was no adjustment for the case mix. The aim of the present study was to quantify excess mortality after single and repeated admissions to a PICU, compared with an age and sex matched control group from the general Swedish population over time. The study group was characterized by age, sex, severity of illness and PICU admission diagnosis.

## Materials and methods

### Setting

This study was conducted at the PICU at the Astrid Lindgren Children's Hospital, Karolinska University Hospital, Stockholm, Sweden. This tertiary referral center for critically ill children has roughly 750 annual admissions and provides intensive care to medical and surgical patients, including neurosurgery, thoracic surgery, and extracorporeal membrane oxygenation

(ECMO) therapy but not cardiac surgery. Non-elective admissions make up approximately 80% of the total admissions to the PICU.

## Data source

The study was conducted as a single-center, retrospective study of prospectively collected data. It was approved by the Swedish Ethical Review Authority (DNr 2013/1078-31/2, 2016/1789-32 and 2020–05528) and registered in the Australian New Zealand Clinical Trials Registry with registration number ACTRN12621001303831. The data contained no protected health information, and there were no changes in patient care due to database entry, thus the need for informed patient consent was waived. Data were collected from consecutive admissions to the PICU from January 1, 2008, to December 31, 2016. The study population consisted of neonatal patients aged from >36 gestational weeks to adolescents aged 18 years. Patients admitted to the PICU are registered in a Patient Data Management System (PDMS; Centricity Critical Care; GE Healthcare Sverige AB, Danderyd, Sweden). The Pediatric Index of Mortality (PIM) score and its different variables are documented in the database. PIM is a widely used model to predict intensive care mortality (predicted death rates [PDRs]) for children by using admission data [19, 20]. The latest version of the PIM score (PIM-3) was adopted in Sweden at the end of 2016; thus, only PIM-2 was used in the present study.

## Data collection and patients

All data from the PDMS were received in electronic form and transformed into a Microsoft Excel 2010 spreadsheet (Microsoft, Redmond, WA). All entered data were manually and electronically checked for errors, that is, extreme or obviously conflicting data. The subjects were divided into four different age groups (0–28 d, > 28 d–1 yr, > 1–4 yr, and > 4 yr) and four different risk stratification groups (PDR intervals: 0–10%, > 10–25%, >25–50%, and > 50%) at admissions. All admissions were investigated regarding background data, PICU admission diagnosis (International Classification of Diseases, 10th Edition, Swedish Edition), and PICU length of stay (PICU-LOS). Mortality included discontinuation of treatment. PICU mortality was gathered from the PDMS for all patients. Data on survival status after PICU discharge were obtained from the population register (Swedish Tax Agency), a separate national database, in which the Swedish population is registered with a unique personal identification number that follows each Swedish citizen from birth to death. The control population consisted of all Swedish individuals of the same age and sex. Statistics Sweden (Statistiska Centralbyrån, Statistikmyndigheten SCB; www.scb.se) provided the mortality data regarding the children in the control group.

A complete dataset on the entire study cohort is available as supplementary material (S1 Table).

## Statistical analysis

Data are reported as n (%) and median values (IQR) unless otherwise stated, where n is the number of observations and IQR is interquartile range (IQR). Survival was evaluated by Kaplan-Meier curves. Statistics were evaluated by MS Excel (Microsoft, Redmond, WA) and GraphPad Prism version 5.04 (Graph Pad Software, San Diego, CA). Significance was defined as $p < 0.05$. Reported p values are from two-sided tests.

## Estimation of standardized mortality ratio (SMR)

The SMRs were calculated using the age and sex matched Swedish population as a reference group. For each patient we used the age at admission and follow-up time, or time to death, in

weeks. Tables with death rates from the Swedish general population classified by age and sex were received from Statistics Sweden [21]. SMR with 95% confidence intervals was calculated from observed mortality of the PICU population with matched median values of death/week per 1,000 controls from the period 2008–2016 in the general population.

The observed mortality compared to the reference population, expressed as SMR, was evaluated according to the principles given in [22–24].

SMR was evaluated for age and risk stratification groups and presented in funnel plots.

## Results

In total, 6,671 admissions were registered in the PICU during the period 2008–2016. We excluded 184 admissions: 124 due to the lack of a personal Swedish identification number, 25 due to a protected identity and 53 because of missing PDR information at admission. Survival status was traced in all 6487 remaining admissions (4,682 individual patients; 2,699 males (57.6%)). Readmission to the PICU were seen in 929 patients (19.8%). Of readmitted patients, 556 patients (59.8%) had two PICU admissions. The study participants are depicted in a Consort flow diagram, Fig 1. The hierarchical nature of patients with repeated admissions to PICU is presented as supplementary material (S2 Table).

Patient characteristics and diagnostic groups based on the reason for admission to PICU, by age groups for single and repeated admissions, are presented in Tables 1 and 2 (first admission) and Table 3 (last admission), respectively.

### Survival and sex after single and repeated admissions to PICU

The SMR for all patients admitted to the PICU during the study period was 49.8 (95%CI 44.8–55.4). There was a statistically significant difference in survival for patients with a single admission and patients with subsequent repeated admissions to PICU (p<0.0001), Fig 2. No significant difference was found in survival between boys and girls neither with a single admission (p = 0.57), nor with repeated admissions to PICU (p = 0.22), Fig 2. The number of patients at follow-up years 0 to 12 for the entire study cohort is presented as supplementary material (S3 Table). After 4 years, the survival rate was 0.942 and 0.936 for boys and girls with a single admission to PICU, respectively. For patients with repeated admissions to PICU the four-year survival rate was 0.847 for boys and 0.822 for girls, respectively. Of all deaths (n = 444) in the study cohort, 30.0% (n = 133) died in PICU. PICU death rate for the entire study cohort was 2.84%, and none-PICU death rate was 6.64%. Death rate during the first year after PICU discharge for the entire study cohort was 4.25% (n = 199).

### SMR and follow-up time after single and repeated admissions to PICU

SMRs at different time periods after PICU admission are shown in Fig 3. The SMRs for the follow up period from PICU admission (= 0 years) until the end of the observation period, i.e., up to 12.5 years, for the PICU patient population with single and repeated admissions was 35.2 (95% CI: 30.5–40.6) and 108.0 (95% CI: 91.9–126.9), respectively. At the 1-year follow up, i.e., one year after PICU admission to the end of the observation period, the SMRs had decreased to 18.2 (95% CI:13.7–24.0) for single and 65.5 (95% CI: 50.9–70.2) for repeated admissions, respectively. The SMRs were 11.0 (95% CI: 7.0–17.6) and 33.9 (95% CI: 23.9–48.0) at the 4-year follow-up, i.e., four years after PICU admission to the end of the observation period, for single and repeated admissions respectively. Of all deaths (n = 444) in the study cohort, 30.9% (n = 137) died in PICU. Subsequent mortality during the first year after PICU discharge was 46.0% (n = 204).

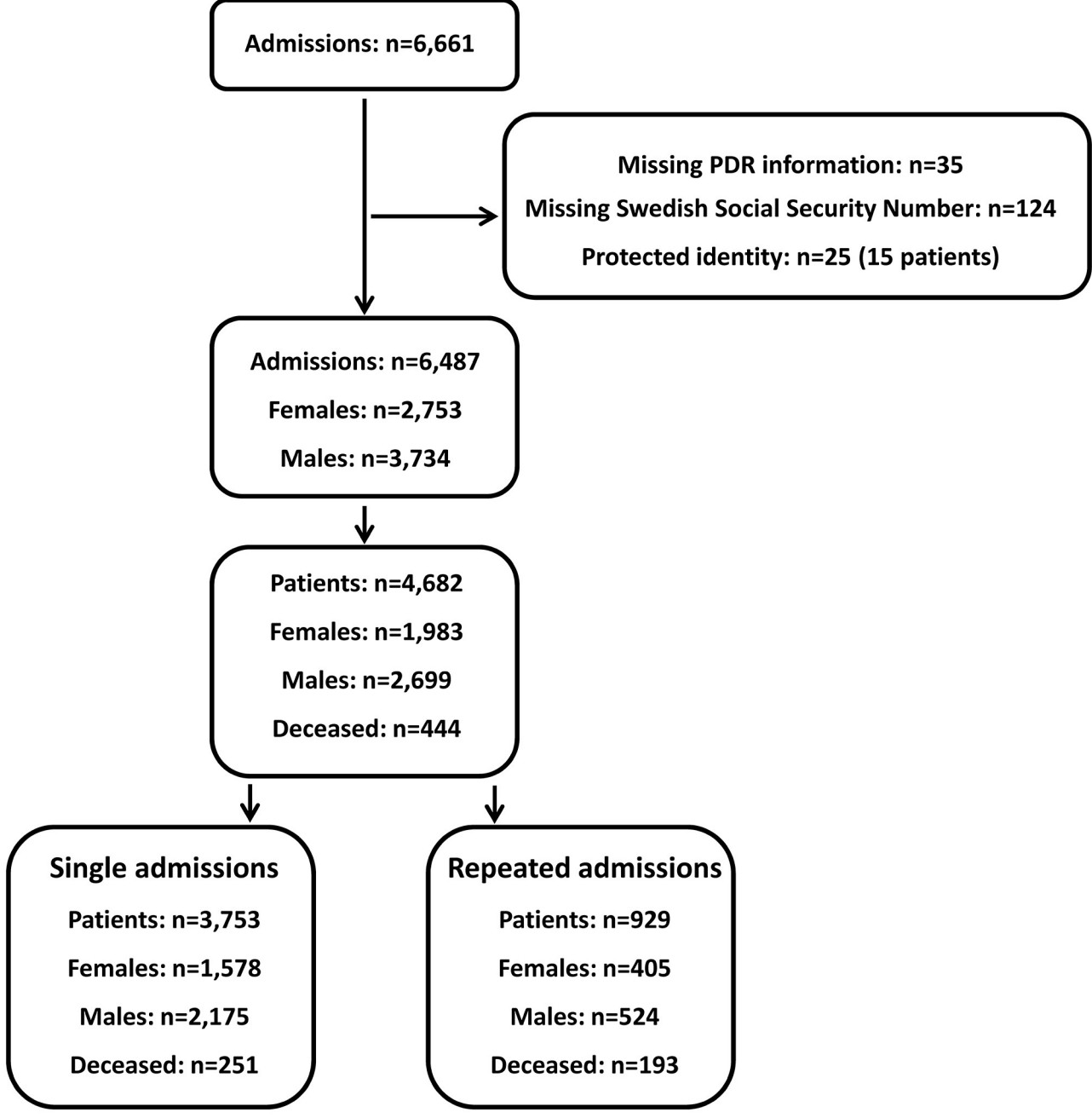

**Fig 1. Consort diagram.** The cohort of patients included in the study, derived from all admissions to the PICU, and presented as single and repeated admissions.

### SMR, age and risk stratification groups

In patients with a single admission there were 3,400 children with a PDR of 0–10%. In this PDR group the SMR of 22.0 was outside the lower 99.8% confidence limit of SMR for the entire study group. In all other PDR groups SMR was outside the upper 99.8% confidence limit. Patients aged 0–28 d with a single admission (n = 779), had a SMR of 17.0 which was

**Table 1. Patient characteristics and PICU admission diagnosis by age groups: Patients with a single admission.**

| | Age group | | | | |
|---|---|---|---|---|---|
| | **0–28 days** | **>28 days—1 year** | **> 1–4 years** | **> 4 years** | **All patients** |
| Number of patients, n (%) | 771 (20.50) | 945 (25.20) | 728 (19.40) | 1309 (34.90) | 3753 |
| Number of boys, n (%) | 458 (59.40) | 585 (61.90) | 418 (57.40) | 714 (54.50) | 2175 (58.00) |
| Age, median (IQR) | 5.75 days | 96.90 days | 1.95 years | 10.70 years | 1.35 years |
| | (1.40–15.50) | (55.00–197.40) | (1.39–2.79) | (6.94–14.00) | (0.12–7.39) |
| PDR, median (IQR) | 2.08 | 1.18 | 1.31 | 1.05 | 1.30 |
| | (1.02–6.30) | (0.58–3.88) | (0.75–4.26) | (0.75–3.16) | (0.75–3.89) |
| Care time days, median (IQR) | 2.40 | 1.17 | 0.86 | 0.81 | 1.00 |
| | (0.97–4.84) | (0.76–2.99) | (0.54–1.92) | (0.52–1.53) | (0.64–2.75) |
| **Diagnostic groups, n (%)** | | | | | |
| Nephrology | 8 (1.04) | 10 (1.06) | 6 (0.82) | 15 (1.15) | 39 (1.04) |
| Certain perinatal conditions | 94 (12.20) | 40 (4.23) | 2 (0.27) | 0 (0) | 136 (3.62) |
| Gastrointestinal including hepatic failure | 41 (5.32) | 41 (4.34) | 25 (3.43) | 39 (2.98) | 146 (3.89) |
| Infection including septic shock + systemic inflammatory response syndrome | 43 (5.58) | 54 (5.71) | 48 (6.59) | 62 (4.74) | 207 (5.52) |
| Haematology and oncology | 12 (1.56) | 27 (2.86) | 67 (9.20) | 94 (7.18) | 200 (5.33) |
| Cardiovascular/Circulatory | 53 (6.87) | 51 (5.40) | 28 (3.85) | 86 (6.57) | 218 (5.81) |
| Endocrine and metabolic diseases | 14 (1.82) | 44 (4.66) | 50 (6.87) | 195 (14.90) | 303 (8.07) |
| Trauma and poison | 6 (0.78) | 41 (4.34) | 113 (15.50) | 368 (28.10) | 528 (14.10) |
| Observations, postoperative or symptom based | 28 (3.63) | 116 (12.30) | 45 (6.18) | 162 (12.40) | 351 (9.35) |
| Neurological including convulsions | 15 (1.95) | 67 (7.09) | 120 (16.50) | 144 (11.00) | 346 (9.22) |
| Malformations | 288 (37.40) | 118 (12.50) | 22 (3.02) | 29 (2.22) | 457 (12.20) |
| Respiratory | 165 (21.40) | 336 (35.60) | 202 (27.80) | 113 (8.63) | 816 (21.70) |
| Unknown | 4 (0.52) | 0 (0) | 0 (0) | 2 (0.15) | 6 (0.16) |

IQR = inter quartile range. Diagnostic groups based on the reason for PICU admission.

outside the lower 99.8% confidence limit. Children aged >1–4 years with a single admission (n = 728) had a SMR of 74.7 which was outside the upper 99.8% confidence limit, Fig 4A.

For patients with repeated admissions to PICU, the PDR group 0–10% (n = 798) had a SMR of 100 at their first admission. PDR groups 10–25% and 25–50% were outside the upper 95% confidence limits. PDR group > 50% was within the 95% confidence limits of SMR for the study group. Neonatal children aged 0–28 d, with repeated admissions (n = 229) had a SMR of 51.4 which was outside the lower 99.8% confidence limit. Children aged >1–4 years had a SMR of 325 which was outside the upper 99.8% confidence limit, Fig 4B.

## SMR and diagnostic groups

The SMR for the various PICU admission diagnosis, compared to the general population, both for patients with a single admission and at the first admission in patients with repeated PICU admissions is illustrated in Fig 5. In all diagnostic groups, with exception for the Cardiovascular/Circulatory group, the SMR was higher in patients with repeated admissions at their first admission as compared to patients with a single admission.

In patients with a single admission the SMR showed a 191-fold increased mortality rate in patients admitted due to cardiovascular/circulatory diagnosis. The second highest SMR was found in the group of patients with oncology/hematology diagnosis (SMR 179). In the groups of patients with diagnosis of Malformation, Trauma/Poison and Certain perinatal conditions the SMR showed the lowest mortality rates (17.9, 18.6 and 20.5), respectively.

**Table 2. Patient characteristics and PICU admission diagnosis by age groups: Patients with repeated admissions: First admission.**

| | Age group | | | | |
| | 0–28 days | >28 days—1 year | > 1–4 years | > 4 years | All patients |
|---|---|---|---|---|---|
| Number of patients, n (%) | 229 (24.70) | 312 (33.60) | 152 (16.40) | 236 (25.40) | 929 |
| Number of boys, n (%) | 131 (57.20) | 189 (60.60) | 97 (63.80) | 107 (45.30) | 524 (56.40) |
| Age, median (IQR) | 1.76 days | 113 days | 1.88 years | 9.56 years | 0.50 years |
| | (0.68–10.80) | (68.20–180.00) | (1.36–2.64) | (6.47–13.10) | (0.08–4.14) |
| PDR, median (IQR) | 2.27 | 1.59 | 1.28 | 1.14 | 1.55 |
| | (0.92–6.57) | (0.75–4.71) | (0.75–4.92) | (0.75–4.04) | (0.75–4.91) |
| Care time days, median (IQR) | 1.98 | 1.76 | 1.02 | 0.92 | 1.28 |
| | (0.79–6.94) | (0.85–4.59) | (0.60–2.15) | (0.61–2.58) | (0.73–4.18) |
| **Diagnostic groups, n (%)** | | | | | |
| Nephrology | 3 (1.31) | 7 (2.24) | 3 (1.97) | 1 (0.42) | 14 (1.51) |
| Certain perinatal conditions | 8 (3.49) | 15 (4.81) | 1 (0.66) | 1 (0.42) | 25 (2.69) |
| Gastrointestinal including hepatic failure | 14 (6.11) | 15 (4.81) | 6 (3.95) | 11 (4.66) | 46 (4.95) |
| Infection including septic shock + systemic inflammatory response syndrome | 8 (3.49) | 16 (5.13) | 8 (5.26) | 9 (3.81) | 41 (4.41) |
| Haematology and oncology | 4 (1.75) | 15 (4.81) | 15 (9.87) | 22 (9.32) | 56 (6.03) |
| Cardiovascular/Circulatory | 15 (6.55) | 22 (7.05) | 5 (3.29) | 21 (8.90) | 63 (6.78) |
| Endocrine and metabolic diseases | 7 (3.06) | 13 (4.17) | 5 (3.29) | 14 (5.93) | 39 (4.20) |
| Trauma and poison | 0 (0) | 5 (1.60) | 6 (3.95) | 22 (9.32) | 33 (3.55) |
| Observations, postoperative or symptom based | 8 (3.49) | 25 (8.01) | 17 (11.20) | 34 (14.40) | 84 (9.04) |
| Neurological including convulsions | 3 (1.31) | 37 (11.90) | 29 (19.10) | 55 (23.30) | 124 (13.30) |
| Malformations | 126 (55.00) | 40 (12.80) | 6 (3.95) | 5 (2.12) | 177 (19.10) |
| Respiratory | 32 (14.00) | 102 (32.70) | 51 (35.60) | 41 (17.40) | 226 (24.30) |
| Unknown | 1 (0.44) | 0 (0) | 0 (0) | 0 (0) | 1 (0.11) |

IQR = inter quartile range, Diagnostic groups based on the reason for PICU admission.

The SMR showed a 358-fold increased mortality rate in patients admitted due to oncology/hematology diagnoses, at the first admission, in patients with repeated PICU admissions. The second highest SMR (192) was found in the group admitted due to neurology (including convulsions) diagnosis. In the groups of patients with Trauma/Poison and Malformation as causes for admission the SMR showed the lowest mortality rate (36.1 and 57.7), respectively.

## Discussion

To our knowledge this is the first study which has compared the mortality of a PICU cohort, using data on the severity of illness affecting the study cohort, to the mortality of the general population of the same age and sex. The study has an extended longitudinal follow-up, of at least four but up to twelve years, which is well beyond the typical duration of study follow-up in intensive care.

There was a significant mortality during the first year after admission to PICU in the entire patient population as reflected by the decrease in survival rate seen in Fig 2. Nearly 50% of all deaths occurred during the first year after PICU discharge, indicating a clinically significant delayed mortality. During the following years only a minor decrease in survival was seen in patients with a single admission. In contrast, there was a continuing decrease in survival during the entire observation period for patients with repeated admissions to PICU.

No statistically significant difference in survival was found between boys or girls, irrespective if they had a single admission to PICU, or if they were repeatedly admitted, Fig 2. These

**Table 3. Patient characteristics and PICU admission diagnosis by age groups: Patients with repeated admissions: Last admission.**

| | Age group | | | | |
|---|---|---|---|---|---|
| | **0–28 days** | **>28 days—1 year** | **> 1–4 years** | **> 4 years** | **All patients** |
| Number of patients, n (%) | 75 (8.10) | 313 (33.70) | 222 (23.90) | 319 (34.30) | 929 |
| Number of boys, n (%) | 40 (53.30) | 197 (62.90) | 126 (56.80) | 161 (50.50) | 524 (56.40) |
| Age, median (IQR) | 13.70 days | 126.60 days | 2.07 years | 9.19 years | 1.66 years |
| | (5.98–18.40) | (66.80–209.00) | (1.50–2.83) | (6.09–13.40) | (0.35–6.30) |
| PDR, median (IQR) | 1.46 | 1.30 | 1.17 | 1.07 | 1.17 |
| | (0.75–3.81) | (0.71–3.91) | (0.59–3.64) | (0.75–3.88) | (0.75–3.88) |
| Care time days, median (IQR) | 1.76 | 1.67 | 1.01 | 1.10 | 1.26 |
| | (0.90–3.79) | (0.85–4.26) | (0.64–3.22) | (0.61–3.18) | (0.76–3.65) |
| **Diagnostic groups, n (%)** | | | | | |
| Nephrology | 1 (1.33) | 5 (1.60) | 7 (3.15) | 4 (1.25) | 17 (1.83) |
| Certain perinatal conditions | 3 (4.00) | 13 (4.15) | 1 (0.45) | 1 (0.31) | 18 (1.94) |
| Gastrointestinal including hepatic failure | 7 (9.33) | 26 (8.31) | 6 (2.70) | 15 (4.70) | 54 (5.81) |
| Infection including septic shock + systemic inflammatory response syndrome | 1 (1.33) | 12 (3.83) | 16 (7.21) | 11 (3.45) | 40 (4.31) |
| Haematology and oncology | 1 (1.33) | 14 (4.47) | 17 (7.66) | 24 (7.52) | 56 (6.03) |
| Cardiovascular/Circulatory | 4 (5.33) | 19 (6.07) | 11 (4.95) | 23 (7.21) | 57 (6.14) |
| Endocrine and metabolic diseases | 2 (2.67) | 13 (4.15) | 8 (3.60) | 20 (6.27) | 43 (4.63) |
| Trauma and poison | 1 (1.33) | 4 (1.28) | 4 (1.80) | 24 (7.52) | 33 (3.55) |
| Observations, postoperative or symptom based | 4 (5.33) | 46 (14.70) | 26 (11.70) | 44 (13.80) | 120 (12.90) |
| Neurological including convulsions | 1 (1.33) | 24 (7.67) | 39 (17.60) | 77 (24.10) | 141 (15.20) |
| Malformations | 30 (40.00) | 46 (14.70) | 17 (7.66) | 4 (1.25) | 97 (10.40) |
| Respiratory | 20 (26.70) | 91 (29.10) | 69 (31.10) | 72 (22.60) | 252 (27.10) |
| Unknown | 0 (0) | 0 (0) | 1 (0.45) | 0 (0) | 1 (0.11) |

IQR = inter quartile range. Diagnostic groups based on the reason for PICU admission.

results are unlike reports from previous studies, where survival analysis have shown an advantage for boys [25–27].

The increased mortality rate found in the present study is much higher than in previous reports from both Finland and Sweden where corresponding risks compared to the general population have been reported to be six-fold and 20-fold greater, respectively [17, 18]. In the present study PICU patients were separated into single and repeated admissions which partly can explain this difference. For patients with a single admission, the SMR showed an 11-fold increased mortality rate compared to the general population four years after PICU admission Fig 3, which is still higher than in the Finnish study, but could be explained by a high rate of postoperative admissions with length of stay in PICU less than 24 hours in that report [18].

The highest SMR was found in the age group > 1–4 years, and the lowest was found in the neonatal group, Fig 4. This was regardless of children being admitted one or repeated times to PICU. Neonatal children in this study may have lower than expected SMRs because the children were not managed in a NICU and were no more premature than 36 weeks. Even though they still had elevated SMR compared to the general population, this may have inflated their survivability compared to the remainder of the PICU population, causing the SMR to be below the 99.8% confidence limit. Additionally, a neonate <28 days old with multiple PICU admissions nearly by definition survived the neonatal period.

Children >1–4 years with repeated admissions had a SMR which showed a more than 300-fold increased mortality rate, compared with the general population, at their first admission. Oncology/hematology and neurologic diagnosis at admission accounted for nearly 30%

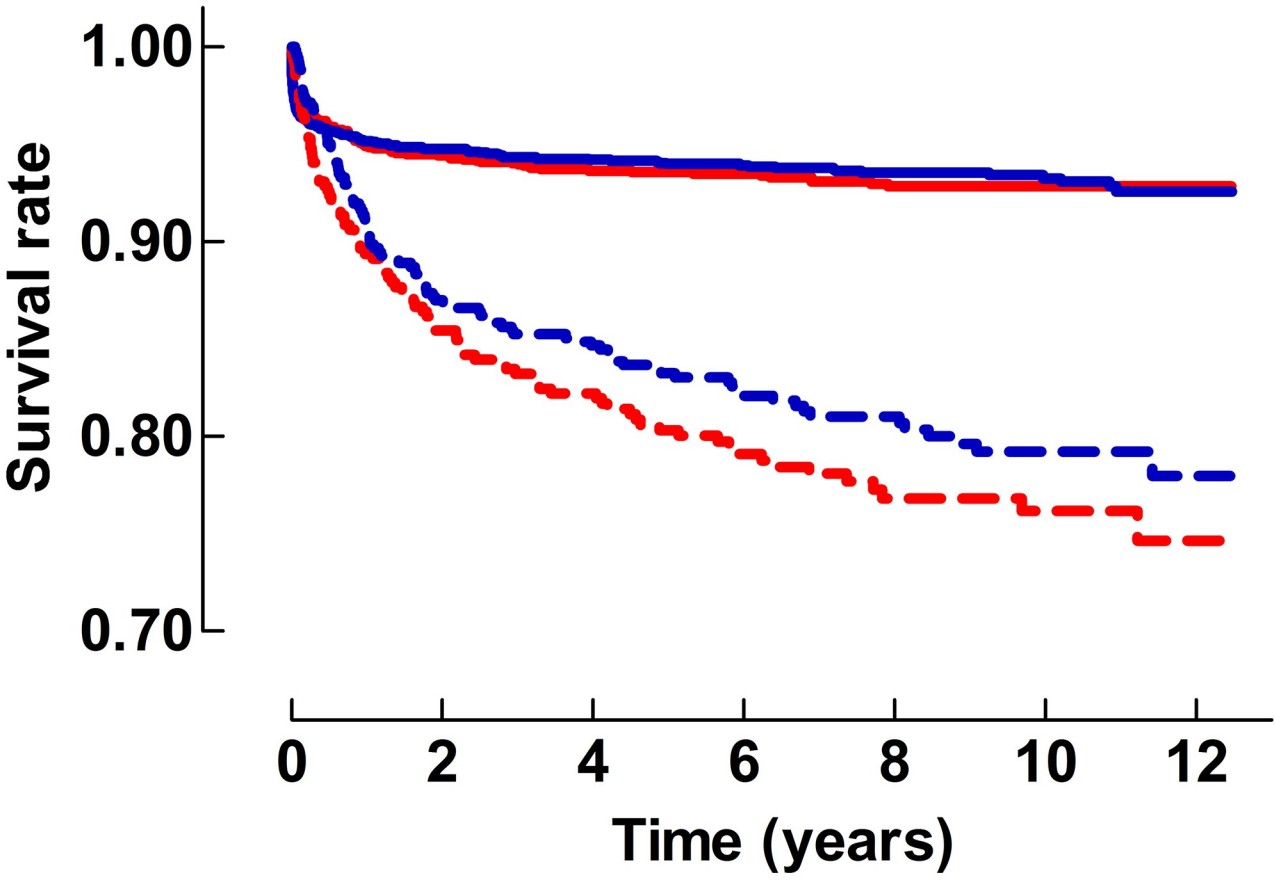

**Fig 2. Survival rates from the first admission to the PICU.** The figure shows a comparison of the survival rate with time from the first admission to the PICU after single and repeated admissions respectively. Solid lines = Data from patients with a single admission. Dotted lines: Data from patients with repeated admissions. Blue lines: Data from boys. Red lines: Data from girls.

of PICU admissions in this patient group which might be a contributing factor to the high mortality risk seen in this age group.

The lowest mortality rate was found in risk stratification group 0–10% with a single admission to PICU. Even though the mortality rate for this risk stratification group was lower than the confidence limit in the funnel plot, the group still had a 20-fold increased mortality risk compared to the general population. All other risk stratification groups with a single admission to PICU appeared outside the upper confidence limit, indicating a higher mortality rate compared to the entire study population with single admissions, Fig 4A.

The considerable increased mortality risk seen in this study also included risk stratification group 0–10%. PICU-patients with an acute complication added to complex chronic illness, which is known to account for a considerable part of admissions to PICU [11, 14–16], but not necessarily captured in the PIM-score, could be an explanation.

Oncology/hematology diagnosis had the highest increased mortality rates compared to the general population in the group of patients with repeated admissions to PICU, Fig 5. High increased mortality rates, compared to the general population, in patients with repeated admissions to PICU was also seen in other medical diagnostic groups such as neurology, nephrology, and cardiovascular diseases. As shown in other reports the injury diagnostic

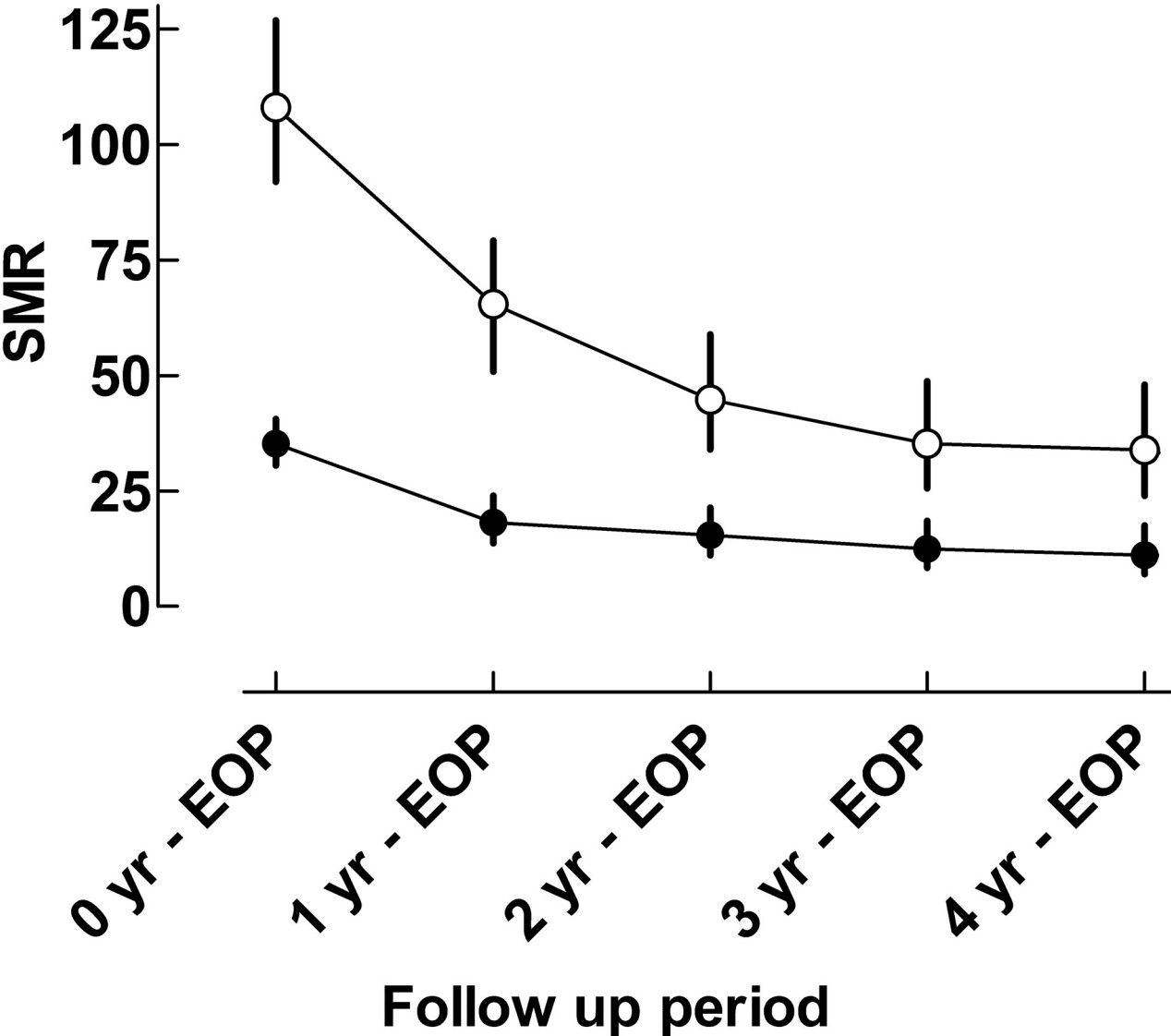

**Fig 3. SMR at different time periods after PICU admission to the end of the observation period.** The figure shows a comparison of SMR from patients with a single admission (Closed symbols) and repeated admissions (Open symbols). SMR expresses the mortality risk, as compared to the general population of the same age and sex. The 95% confidence intervals are given by the error bars. Statistical significances are indicated by non-overlapping confidence intervals. EOP = End of Observation Period.

group carried a low mortality, compared to the general population [17, 18]. It is expected that underlying complex chronic disease is less frequent in this diagnostic group.

A strength of this study is the linking of PICU data to a national registry to evaluate long-term survival.

Furthermore, consecutive patients were included, they were categorized by diagnoses set by the treating physician in the PICU and not by the researchers involved. The main limitation of the study is that we lack information on the prevalence of complex chronic illness. Our ambition is to collect information on the presence of complex chronic conditions in a follow-up study. Data on long-term morbidity and quality of life, which are important outcome factors after critical illness and pediatric intensive care, were also not possible to obtain. To illuminate

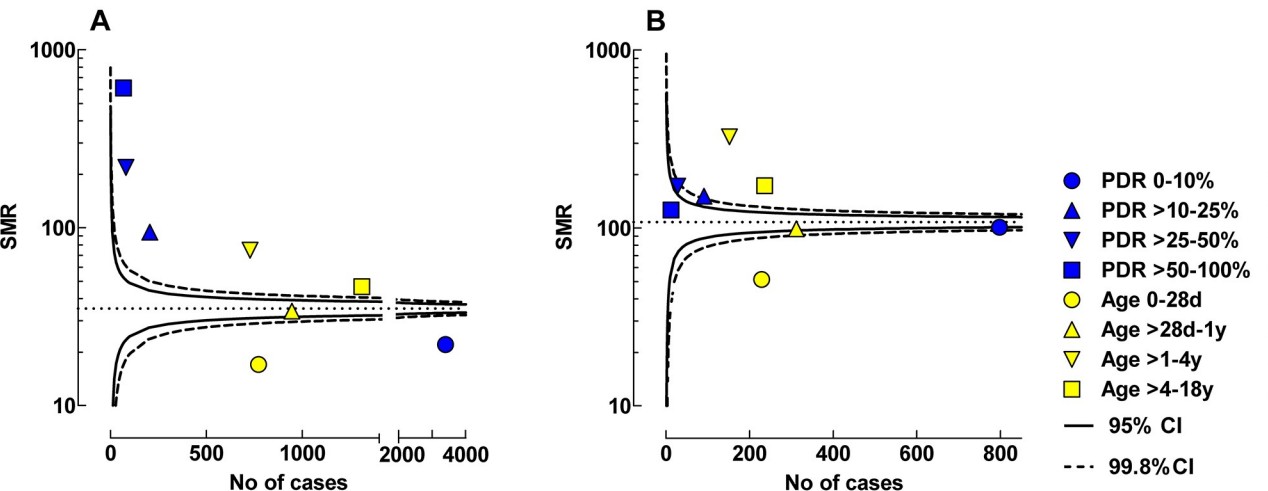

**Fig 4. Funnel plots showing SMR for different age and risk stratification groups.** The presented data are from the first admission to the PICU. A: Single admission to PICU. B: Repeated admissions to PICU. The 95 and 99.8% confidence intervals of SMR of the entire study population, in relation to number of cases, are given by the solid and dotted lines, respectively.

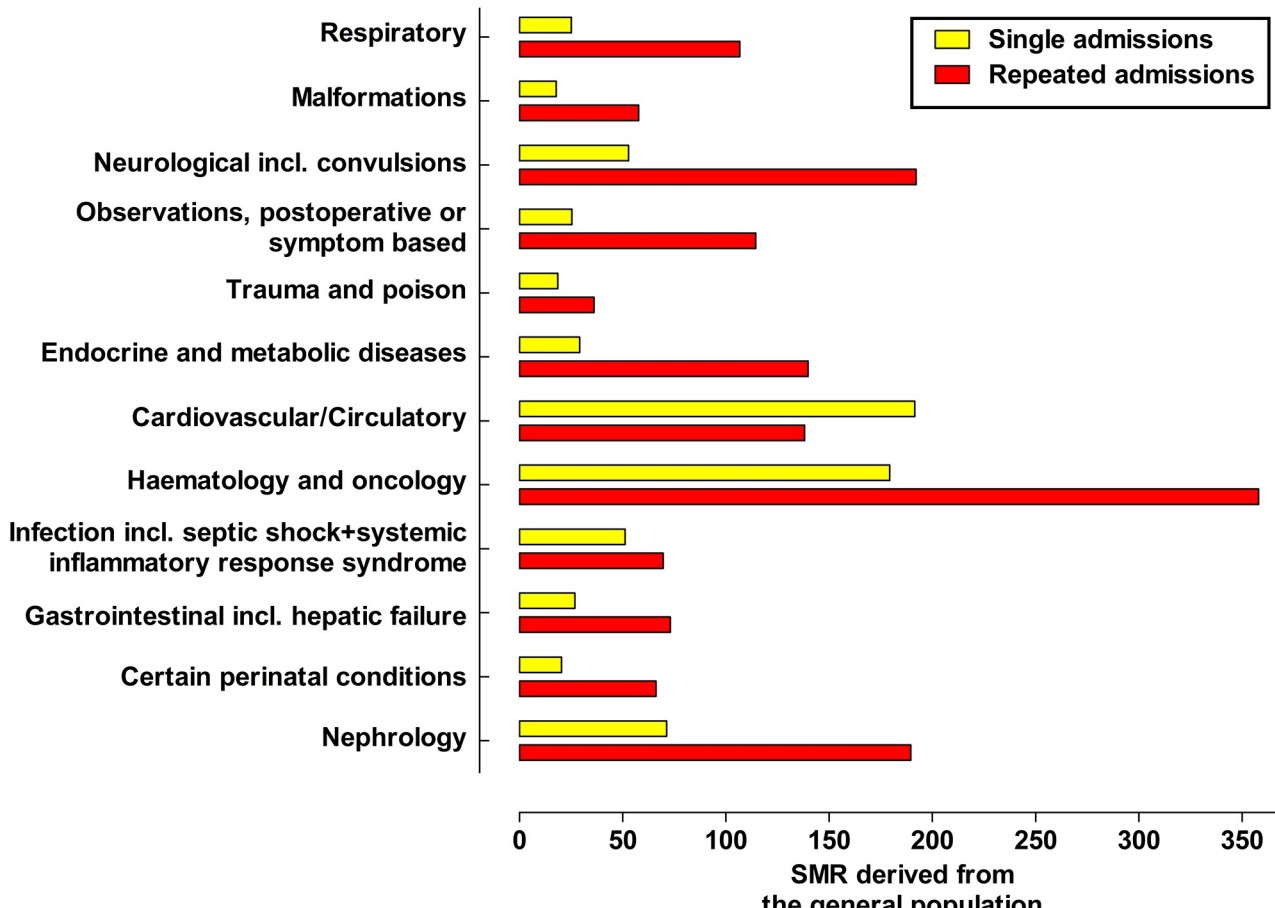

**Fig 5. Bar graph showing SMR for different PICU admission diagnosis.** The presented data are from the first admission to the PICU. The SMRs are derived from the general population.

questions around chronic illness patients were stratified into single versus repeated admissions. Multiple PICU admissions likely functions as a surrogate marker for chronic illness, accounting for the increase mortality risk in these patients. This is further supported by the lower number of "repeat" patients with trauma/poisoning diagnoses. Third, although our study contains data over twelve years, a limitation is that data originates from a single center, and thereby limiting generalizability.

Finally, we were not able to remove the study group from the matched general population, since we were not able to separate this group in the national registers. If any effect at all, this means that the calculated SMRs could be expected to be even higher if the study group had been excluded from the matched general population.

## Conclusion

Compared to the age and sex matched Swedish population, the SMR for all patients admitted to the PICU during the study period was nearly 50. SMRs in general were three times higher for patients with repeated admissions to PICU compared to single admitted patients. There was significant excess mortality over time in PICU survivors. SMRs were greatly elevated up to four years after PICU admission, declining from over 100 to 33 for patients with repeated PICU admissions, and from 35 to 11 for patients with a single PICU admission.

## Supporting information

**S1 Table. Complete dataset on the entire study population.** Anonymized.
(PDF)

**S2 Table. Number of repeated admissions.** The hierarchical nature of patients with repeated admissions to PICU.
(PDF)

**S3 Table. Patients at risk.** The number of patients at follow-up years 0 to 12 for the entire study cohort.
(PDF)

## Author Contributions

**Conceptualization:** Tova Hannegård Hamrin, Staffan Eksborg.

**Data curation:** Staffan Eksborg.

**Formal analysis:** Tova Hannegård Hamrin, Staffan Eksborg.

**Investigation:** Tova Hannegård Hamrin, Staffan Eksborg.

**Methodology:** Tova Hannegård Hamrin, Staffan Eksborg.

**Project administration:** Tova Hannegård Hamrin.

**Software:** Staffan Eksborg.

**Validation:** Tova Hannegård Hamrin, Staffan Eksborg.

**Visualization:** Staffan Eksborg.

**Writing – original draft:** Tova Hannegård Hamrin.

**Writing – review & editing:** Tova Hannegård Hamrin, Staffan Eksborg.

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
