## [Decision Letter · Decision Letter 0]

18 Apr 2022

PONE-D-22-06716Risks for death after admission to pedaitric intensive care (PICU) - a comparison with the general populationPLOS ONE

Dear Dr. Hannegård Hamrin,

Thank you for submitting your manuscript to PLOS ONE. After careful consideration, we feel that it has merit but does not fully meet PLOS ONE’s publication criteria as it currently stands. Therefore, we invite you to submit a revised version of the manuscript that addresses the points raised during the review process.

We look forward to receiving your revised manuscript.

Kind regards,

Brenda M. Morrow, PhD

Academic Editor

PLOS ONE

Journal Requirements:

Reviewers' comments:

Reviewer's Responses to Questions

**Comments to the Author**

1. Is the manuscript technically sound, and do the data support the conclusions?

Reviewer #1: Partly

Reviewer #2: Yes

2. Has the statistical analysis been performed appropriately and rigorously? 

Reviewer #1: No

Reviewer #2: Yes

3. Have the authors made all data underlying the findings in their manuscript fully available?

Reviewer #1: No

Reviewer #2: No

4. Is the manuscript presented in an intelligible fashion and written in standard English?

Reviewer #1: Yes

Reviewer #2: Yes

5. Review Comments to the Author

Reviewer #1: The authors have utilised a large population based dataset to compare mortality in PICU patients in PICU or after discharge and have compared this, suing relative measure, to the general population of children. Given that children are admitted to PICU with life-threatening medical or surgical condition it is unsurprising that their SMR is markedly higher that the general population. I am also unclear how valuable using relative measures of risk alone are in this situation - in fact could cause great concern amongst parents. I think further re-analyses and clarification of the importance of these findings i.e "so what' are important before this paper is suitable for publication.

Major concerns

1. please present absolute risks as well as relative risks throughout the manuscript

2. please clearly articulate the 'so what' of these analyses

3. Is the diagnostic group based on reason for admission to PICU or underlying health conditions? Most children admitted to PICU have underlying complex medical condition so understanding this groups is very important when thinking through the implications of this study.

4. I would ask that the authors present the data for within PICU and after PICU discharge mortality see this paper https://adc.bmj.com/content/103/6/540. Understanding the risk after discharge is really important in terms of needs for palliative care services and parental expectations

5. What about children with multiple diagnoses, they often have higher risk of mortality?

6. I am unclear how the hierarchical nature of these data have been accounted for in these analyses - ie. children with multiple admissions?

7. Please present the numbers of children where life-sustaining treatment was withdrawn within ICU

8. all these comments relate to being able to identify (prospectively) which children may be at higher risk of death- doing this retrospectively once they have had more than one PICU admission is not necessarily that useful

Minor points

1. two decimal places are adequate in tables

2. this sentence is incorrect "Subsequent mortality during the first year

176 after PICU discharge was 46.0% (n=204)" and should be corrected

Reviewer #2: Thank you for the opportunity to review this paper, which compared the mortality of PICU-admitted children to those of the matched general Swedish population.

The comparison was further stratified into various categories: age, risk factor (via PIM2), and diagnosis. Importantly, splitting patients according to whether they had only a single PICU admission or multiple ones was appropriate because the authors’ premise about multiple PICU admissions serving as a proxy for chronic illness/conditions is a sound one.

Main comment:

My biggest issue is the lack of granularity in the data provided in the submission. Having the raw numbers of what went into the SMR calculations would be helpful; at the very least – the observed deaths of the PICU patients at the follow-up times, in the binned categories, etc). Would it be possible to provide this as Supplementary Material? The SMRs shown in Figures 3-5 can only be eye-balled and cannot be reproduced from the data provided in the submission. This is why I answered “No” to Question 3 (regarding data underlying findings in the manuscript).

These questions/comments are to help clarify details of computations and Figures:

a) Figure 2: Assuming that the study year was 2020, the number of total patients (needed in the denominator) after year 4 is steadily decreasing – e.g. all admissions (years 2008-2016) are eligible for Time<4 years, but only those admitted in 2008 are available at Time=12 years (after admission). It might be helpful to annotate in the Figure (or provide in a supplementary table) the number of patients at each time point (years after admission) that the survival rate is being plotted.

b) Figure 3 and lines 169-171 either show or mention SMR at “PICU admission” – this may be a misunderstanding on my part, but how do you get a non-zero SMR (i.e. non-zero observed deaths) at time of admission? For this, are you counting the deaths that occurred in the PICU at some later time than admission?

c) On page 14, line 191 states “All PDR groups were within the 95% CI limits of SMR …” – this is for repeated admissions; however in Figure 4B, the SMRs for 2 PDR groups (10-25%, 25-50%) look just a little outside the limits.

Other (minor) comments:

i) Full title in the submission (not manuscript) – “pedaitric” should be “pediatric”

ii) Lines 227-228 (Discussion section) – Figure 2 (the survival rate) is a stronger support for this statement than the 1-year SMR.

6. PLOS authors have the option to publish the peer review history of their article (what does this mean?). If published, this will include your full peer review and any attached files.

Reviewer #1: **Yes: **Professor Lorna Fraser

Reviewer #2: No

---

## [Author Response · Author response to Decision Letter 0]

11 May 2022

Responses to Reviewers' Comments to Author,

Thank-you to both Reviewers for your constructive and thoughtful comments and suggestions. We are grateful for the opportunity to revise our article and agree that the manuscript can be and has been much improved by changes according to your suggestions.

Below, are our responses to your suggestions and comments. 

Reviewer #1: The authors have utilised a large population based dataset to compare mortality in PICU patients in PICU or after discharge and have compared this, suing relative measure, to the general population of children. Given that children are admitted to PICU with life-threatening medical or surgical condition it is unsurprising that their SMR is markedly higher that the general population. I am also unclear how valuable using relative measures of risk alone are in this situation - in fact could cause great concern amongst parents. I think further re-analyses and clarification of the importance of these findings i.e "so what' are important before this paper is suitable for publication.

Major concerns

1. please present absolute risks as well as relative risks throughout the manuscript

Reply: Thank you for your comment. In this study we have evaluated and reported SMR (standardized mortality ratio) according to the original article by reference 24 in the list of references: Finkelstein DM, Muzikansky A, Schoenfeld DA. Comparing survival of a sample to that of a standard population. J Natl Cancer Inst. 2003;95: 1434-1439. 

When comparing survival of a study cohort with an age and sex matched general population, we think that using the method as described in this reference article is to be preferred. 

The use of SMR and its Confidence intervals as a statistical method is further described by 

Liddell FD: Simple exact analysis of the standardised mortality ratio. J Epidemiol Community Health.

1984; 38:85–88. 

2. please clearly articulate the 'so what' of these analyses

Reply: Thank you for this valuable comment. We agree with the reviewer and think that the conclusion section could be improved and have therefore made a revision. 

3. Is the diagnostic group based on reason for admission to PICU or underlying health conditions? Most children admitted to PICU have underlying complex medical condition so understanding this groups is very important when thinking through the implications of this study.

Reply: Thank you for your question. The diagnostic groups are based on reason for admission to PICU. We have further clarified this in the Material and Methods section page 7, Results section page 9 and in the footnotes of Tables 1-3.

4. I would ask that the authors present the data for within PICU and after PICU discharge mortality see this paper https://adc.bmj.com/content/103/6/540. Understanding the risk after discharge is really important in terms of needs for palliative care services and parental expectations

Reply: Thank you for this suggestion. We have now included results on death rates for the entire study population separated in: PICU death rate, non PICU death rate and death rate during the first year after PICU discharge in the Results section, page 13.

5. What about children with multiple diagnoses, they often have higher risk of mortality?

Reply: Thank you for this comment; we agree with the reviewer that the absence of data on chronic comorbidities is a weakness. To illuminate questions around children with chronic complex illness, patients were stratified into single versus repeated admissions. Multiple PICU admissions likely functions as a surrogate marker for chronic illness, accounting for the increased mortality risk in these patients. The findings are discussed and the lack of data on chronic comorbidities for the entire study cohort are acknowledged in the list of study limitations.

6. I am unclear how the hierarchical nature of these data have been accounted for in these analyses - ie. children with multiple admissions?

Reply: Thank you for your suggestion to clarify this, a table has been included as supplementary material (Suppl. Table S2).

7. Please present the numbers of children where life-sustaining treatment was withdrawn within ICU

Reply: Thank you for your comment. In this study we have not investigated any causes of death, and therefore we are unable to present the numbers of children where life-sustaining treatment was withdrawn within PICU. 

8. all these comments relate to being able to identify (prospectively) which children may be at higher risk of death- doing this retrospectively once they have had more than one PICU admission is not necessarily that useful

Reply: Thank you for your comment. We believe that there might be a small misunderstanding due to lack of clarity in our manuscript and we are grateful for the opportunity to make clarifications.

The aim of the present study was to quantify excess mortality over time in children post admission to a PICU, compared with a control population of same age and sex, ie. the matched general Swedish population. Thus, our aim in this study was not to prospectively identify which children may be at higher risk of death. 

Minor points

1. two decimal places are adequate in tables

Reply: Thank you for your suggestion about the number of decimals. We agree with the reviewer and have changed accordingly in the manuscript.

2. this sentence is incorrect "Subsequent mortality during the first year

176 after PICU discharge was 46.0% (n=204)" and should be corrected

Reply: Thank you for this comment; we have changed the text according to the suggestion. 

Reviewer #2: Thank you for the opportunity to review this paper, which compared the mortality of PICU-admitted children to those of the matched general Swedish population.

The comparison was further stratified into various categories: age, risk factor (via PIM2), and diagnosis. Importantly, splitting patients according to whether they had only a single PICU admission or multiple ones was appropriate because the authors’ premise about multiple PICU admissions serving as a proxy for chronic illness/conditions is a sound one.

Main comment:

My biggest issue is the lack of granularity in the data provided in the submission. Having the raw numbers of what went into the SMR calculations would be helpful; at the very least – the observed deaths of the PICU patients at the follow-up times, in the binned categories, etc). Would it be possible to provide this as Supplementary Material? The SMRs shown in Figures 3-5 can only be eye-balled and cannot be reproduced from the data provided in the submission. This is why I answered “No” to Question 3 (regarding data underlying findings in the manuscript).

Reply: We thank the reviewer for pointing out this important weakness. We are grateful for the opportunity to make clarifications and have now included data on admission year, follow up time, age at admission, sex, outcome (deceased/alive), PDR, and PICU admission diagnosis for the entire study population in Supplementary material as Suppl. Table S1. 

These questions/comments are to help clarify details of computations and Figures:

a) Figure 2: Assuming that the study year was 2020, the number of total patients (needed in the denominator) after year 4 is steadily decreasing – e.g. all admissions (years 2008-2016) are eligible for Time<4 years, but only those admitted in 2008 are available at Time=12 years (after admission). It might be helpful to annotate in the Figure (or provide in a supplementary table) the number of patients at each time point (years after admission) that the survival rate is being plotted.

Reply: Thank you for this valuable comment. We agree that the number of patients at each time point for follow-up should be clarified and have now included a supplementary table (Suppl. Table S3) to give an overview of the number of patients at follow-up years 0 to 12 for the entire study cohort. Patients are presented as single/repeated admissions and female/male admissions.

b) Figure 3 and lines 169-171 either show or mention SMR at “PICU admission” – this may be a misunderstanding on my part, but how do you get a non-zero SMR (i.e. non-zero observed deaths) at time of admission? For this, are you counting the deaths that occurred in the PICU at some later time than admission?

Reply: We appreciate this comment, to clarify we have made changes in the manuscript accordingly, page 14, and have rewritten the text on the x-axis in Figure 3. 

c) On page 14, line 191 states “All PDR groups were within the 95% CI limits of SMR …” – this is for repeated admissions; however in Figure 4B, the SMRs for 2 PDR groups (10-25%, 25-50%) look just a little outside the limits.

Reply: We thank the reviewer for pointing this out to us. We have corrected the text accordingly, page 14-15.

Other (minor) comments:

i) Full title in the submission (not manuscript) – “pedaitric” should be “pediatric”

Reply: Thank you. We have corrected the text accordingly.

ii) Lines 227-228 (Discussion section) – Figure 2 (the survival rate) is a stronger support for this statement than the 1-year SMR.

Reply: We agree and have changed the text according to the suggestion, page 16.

---

## [Decision Letter · Decision Letter 1]

13 Jul 2022

PONE-D-22-06716R1Risks for death after admission to pediatric intensive care (PICU) - a comparison with the general populationPLOS ONE

Dear Dr. Hannegård Hamrin,

Thank you for submitting your manuscript to PLOS ONE. After careful consideration, we feel that it has merit but does not fully meet PLOS ONE’s publication criteria as it currently stands. Therefore, we invite you to submit a revised version of the manuscript that addresses the points raised during the review process.

The reviewer ask for some additional clarifications regarding Figure 3. Could you please revise the manuscript to address these concerns?

We look forward to receiving your revised manuscript.

Kind regards,

Thomas Tischer

Staff Editor

PLOS ONE

Journal Requirements:

Reviewers' comments:

Reviewer's Responses to Questions

**Comments to the Author**

1. If the authors have adequately addressed your comments raised in a previous round of review and you feel that this manuscript is now acceptable for publication, you may indicate that here to bypass the “Comments to the Author” section, enter your conflict of interest statement in the “Confidential to Editor” section, and submit your "Accept" recommendation.

Reviewer #2: (No Response)

2. Is the manuscript technically sound, and do the data support the conclusions?

Reviewer #2: Yes

3. Has the statistical analysis been performed appropriately and rigorously? 

Reviewer #2: Yes

4. Have the authors made all data underlying the findings in their manuscript fully available?

Reviewer #2: Yes

5. Is the manuscript presented in an intelligible fashion and written in standard English?

Reviewer #2: Yes

6. Review Comments to the Author

Reviewer #2: Thank you for providing the supplementary material. The responses, along with the additional tables in the supplements, addressed most of my previous concerns.

I have a remaining comment/question regarding Figure 3. In my previous review, I had asked about the non-zero SMR at year=0 (time of PICU admission). The response did not really clear up my original question.

In Figure 3, there is a certain "binning" taking place for the number of years on the horizontal axis, i.e. data are plotted only for discrete years 0, 1, 2, 3, 4, and 5 after PICU admission. When you plot the SMR for say year=0, are you counting the observed deaths up to, but strictly less than, 1 year from PICU admission? Likewise, does the SMR for year=1 reflect observed deaths that occurred at least 1 year, but strictly less than 2 years, after PICU admission?

This revised statement -- "The SMRs, during the total follow-up time for the entire PICU patient population with single and repeated admissions was 35.2% ... and 108.0 ...." (lines 176-179, page 14) -- where the stated numbers correspond to the year=0 point on the horizontal axis, still does not track with my understanding of what Figure 3 is supposed to be showing. How does "total follow-up time" track with year=0?

Figure 3, especially the number corresponding to year=0 for repeated admissions, is referenced in the Discussion section, so please make sure to explain clearly/thoroughly what year=0 (after PICU admission) really mean.

7. PLOS authors have the option to publish the peer review history of their article (what does this mean?). If published, this will include your full peer review and any attached files.

Reviewer #2: No

---

## [Author Response · Author response to Decision Letter 1]

23 Aug 2022

Thank you for the opportunity to revise our manuscript.

The concern raised by reviewer #2 was that clarifications were needed regarding what Figure 3 is supposed to be showing, especially whether data are plotted only for discrete years and what year=0 after PICU admission means. Significant effort has therefore been put into making a focused effort to improve Figure 3 and its Figure legend. Clarifications have been made accordingly in the result and discussion sections of the manuscript as can be seen in the accompanying “Responses to Reviewer”.

---

## [Decision Letter · Decision Letter 2]

8 Sep 2022

Risks for death after admission to pediatric intensive care (PICU) - a comparison with the general population

PONE-D-22-06716R2

Dear Dr. Hannegård Hamrin,

We’re pleased to inform you that your manuscript has been judged scientifically suitable for publication and will be formally accepted for publication once it meets all outstanding technical requirements.

Kind regards,

Academic Editor

PLOS ONE

Additional Editor Comments (optional):

Reviewers' comments:

Reviewer's Responses to Questions

**Comments to the Author**

1. If the authors have adequately addressed your comments raised in a previous round of review and you feel that this manuscript is now acceptable for publication, you may indicate that here to bypass the “Comments to the Author” section, enter your conflict of interest statement in the “Confidential to Editor” section, and submit your "Accept" recommendation.

Reviewer #2: All comments have been addressed

Reviewer #3: All comments have been addressed

2. Is the manuscript technically sound, and do the data support the conclusions?

Reviewer #2: Yes

Reviewer #3: Yes

3. Has the statistical analysis been performed appropriately and rigorously? 

Reviewer #2: Yes

Reviewer #3: Yes

4. Have the authors made all data underlying the findings in their manuscript fully available?

Reviewer #2: Yes

Reviewer #3: Yes

5. Is the manuscript presented in an intelligible fashion and written in standard English?

Reviewer #2: Yes

Reviewer #3: Yes

6. Review Comments to the Author

Reviewer #2: Thank you for making the clarification about Figure 3.

Reviewer #3: No additional comments from me as I think my previous comments were already addressed in the previous version.

7. PLOS authors have the option to publish the peer review history of their article (what does this mean?). If published, this will include your full peer review and any attached files.

Reviewer #2: No

Reviewer #3: No

---

## [Editor Report · Acceptance letter]

27 Sep 2022

PONE-D-22-06716R2 

Risks for death after admission to pediatric intensive care (PICU) - a comparison with the general population 

Dear Dr. Hannegård Hamrin:

I'm pleased to inform you that your manuscript has been deemed suitable for publication in PLOS ONE. Congratulations! Your manuscript is now with our production department. 

Kind regards, 

on behalf of

Dr. Robert Jeenchen Chen 

Academic Editor

PLOS ONE